# Robust Detection of Background Acoustic Scene in the Presence of Foreground Speech

Siyuan Song * , Yanjue Song and Nilesh Madhu

IDLab, Department of Electronics and Information Systems, Ghent University—imec, 9000 Ghent, Belgium;
yanjue.song@ugent.be (Y.S.); nilesh.madhu@ugent.be (N.M.)
* Correspondence: siyuan.song@ugent.be

**Abstract:** The characterising sound required for the Acoustic Scene Classification (ASC) system is contained in the ambient signal. However, in practice, this is often distorted by e.g., foreground speech of the speakers in the surroundings. Previously, based on the iVector framework, we proposed different strategies to improve the classification accuracy when foreground speech is present. In this paper, we extend these methods to deep-learning (DL)-based ASC systems, for improving foreground speech robustness. ResNet models are proposed as the baseline, in combination with multi-condition training at different signal-to-background ratios (SBRs). For further robustness, we first investigate the noise-floor-based Mel-FilterBank Energies (NF-MFBE) as the input feature of the ResNet model. Next, speech presence information is incorporated within the ASC framework obtained from a speech enhancement (SE) system. As the speech presence information is time-frequency specific, it allows the network to learn to distinguish better between background signal regions and foreground speech. While the proposed modifications improve the performance of ASC systems when foreground speech is dominant, in scenarios with low-level or absent foreground speech, performance is slightly worse. Therefore, as a last consideration, ensemble methods are introduced, to integrate classification scores from different models in a weighted manner. The experimental study systematically validates the contribution of each proposed modification and, for the final system, it is shown that with the proposed input features and meta-learner, the classification accuracy is improved in all tested SBRs. Especially for SBRs of 20 dB, absolute improvements of up to 9% can be obtained.

**Keywords:** acoustic scene classification; speech enhancement; noise floor; deep learning; foreground speech robust ASC; ResNet

## 1. Introduction

Acoustic devices need to be environment-adaptive in order to achieve better user experience. Thus, determining the acoustic environments from acoustic recordings, which is usually termed as acoustic scene classification (ASC), is an advanced research field that many applications can benefit from. For example, for hearing aid devices, different settings/filtering characteristics could be suggested with the changing acoustic scenes based on classifications, like indoor, outdoor, or transportation. Another important application is generating an informative embedding for other algorithms. The background scene information can be embedded in speech enhancement to design tailor-made settings for each scene. Also, for speech separation or speech localisation, the processing can be adapted depending on the types of background scene.

The research on ASC systems started with algorithms based on statistical models [1] which summarize the properties of individual audio scenes or whole soundscape categories, typically including Gaussian mixture models (GMMs) [2], hidden Markov models (HMMs) [3], and iVector [4,5]. GMMs model the statistical distribution of feature vectors associated with different audio scenes. These models represent each acoustic environment as a combination of Gaussian distributions, enabling them to capture diverse

statistical properties inherent in audio features. HMMs specialize in capturing the temporal dynamics and represents acoustic environments as sequences of states, modeling transitions between states over time. This temporal aspect, coupled with the statistical insights provided by GMMs, enhances the ability of ASC systems to discern and classify complex soundscapes accurately. ASC systems that fall into this category normally extract the feature vectors from the training data, and a statistical model representing the characteristics of different classes is then trained using the feature vectors. A decision criterion finally assigns the recordings to the best-matched classes. These hand-engineered systems are modular and highly controllable, which makes it easier to analyse the influence of each module. When ASC datasets become larger and more complex [6,7], deep learning architectures are advantageous. Many DL-based ASC systems are capable of processing large datasets efficiently and show great potential in distinguishing very similar acoustic scenes with high accuracy. Also, they can adapt to complex datasets with recordings from different devices, different geographical locations, and so on. In [8], an overview of DL-based ASC systems is provided. These systems achieved promising results in the Detection and Classification of Acoustic Scenes and Events (DCASE) (https://dcase.community/challenge2021/task-acoustic-scene-classification-results-a) (accessed on 7 November 2023), a re-occurring event that stimulates and leads the development of ASC. Most of the systems utilize convolutional neural networks (CNNs) [9,10] and their extensions. Other architectures include convolutional recurrent neural networks (CRNNs) [11,12], feedforward neural networks (FNNs) [13,14], etc.

The signal characterising the acoustic scene is the background. When only the background signal is present in the recordings, current ASC systems perform well. However, in real-life applications, a significant amount of foreground speech, irrelevant to the acoustic scenes, is also captured, which leads to performance degradation [15]. In this paper, we are interested in improving the ASC robustness in the presence of foreground speech. To our best knowledge, although plenty of research focuses on improving ASC systems in terms of learning architectures and training paradigms, there is very little research on how to improve foreground speech robustness—as we also note in our previous work within the iVector framework [15,16]. A rather naive way would be to remove foreground speech by repurposing existing speech denoising frameworks (e.g., [17]). For example, in [18], removing foreground speech is included as a pre-stage of the ASC framework. Such methods might affect the ASC system by introducing (non-linear) artefacts into the processed signal. Instead, in our previous work, we showed that utilising noise-floor (NF)-based features [19–21] and incorporating speech presence information could improve the foreground speech robustness of the classical iVector-based ASC. It remains to be explored whether DL-based systems could share the benefits of our previous research or whether they can be made inherently robust by training. In this paper, a residual network (ResNet) architecture [22] is taken as the baseline, through which we hope to extend the conclusions also to other deep learning models. This ResNet model, as our entry to the Detection and Classification of Acoustic Scenes and Events (DCASE) 2021 challenge, achieved an accuracy of 68.8% in the evaluation dataset, while the official baseline accuracy was 45.6%. Meanwhile, the performance of the best system in the challenge was 76.1%. ResNet models with different complexity are explored in order to show the generalisation of the proposed methods. We believe the results will extend also to other DL-based frameworks since in this paper we are mainly focusing on optimal input features and ensemble methods.

The noise floor estimate as an input stream in the ASC system is first investigated. The noise floor is obtained by statistical methods and typically yields a highly smoothed estimate of the background signal spectrum. While this feature is less corrupted by the more dynamically varying speech spectrum, the smoothing nevertheless results in loss of information and, subsequent sub-optimal performance of the classifier. To incorporate more signal characteristics in the background dominant frames, therefore, we next consider emphasising these frames by incorporating the speech presence information. For this, a state-of-the-art speech enhancement (SE) system, CRUSE [23], is integrated to provide the

short-time Fourier domain speech mask. This mask can be regarded as the speech presence probability. Finally, for a global decision, ensemble methods including model voting and meta-learner are investigated to achieve the best performance.

The remainder of this paper is organized as follows. In Section 2, we summarize our previous work within the iVector framework, which provides crucial cues to the strategies for deep learning systems. Section 3 presents the ResNet model that is utilized as the baseline, including the model architecture, feature extraction, and training paradigm. Section 4 includes a description of the proposed modifications: use of the noise floor features and the speech presence probability. Different combinations using ensemble methods are introduced for global decisions. Section 5 describes the experimental setup and evaluation results including the different combinations using ensemble methods. Section 6 concludes the paper.

## 2. Previous Work

In previous research, we explicitly investigated methods to improve foreground speech robustness based on a modular and highly controllable iVector framework. In this framework, Mel-Frequency Cepstral Coefficients (MFCCs) are used as the acoustic input features. In the factor analysis stage, the MFCC feature, which extracts the information in the variable-length recording, is compressed to the iVector with a fixed length. In the classification stage, a regularized Gaussian backend classifier is then used to extract the most likely scene by processing the estimated iVector. This forms the baseline.

In [15], we proposed the NF-based iVector framework where the MFCCs are derived from an estimate of the background signal power spectral density. We observed significant accuracy improvement in conditions with high signal-to-background ratios (SBRs), but at the cost of poorer performance with low SBRs (The baseline achieved a better performance in such conditions). Here, the term of SBR is utilized to measure the relation between the speech power and the background signal power in a recording. Note that in the speech enhancement literature this is traditionally termed signal-to-noise ratio, where the background/ambient signal is the "unwanted" component. However, in ASC, it is exactly this signal that we are interested in, and use of the term 'noise' is therefore inappropriate. In [16], the trade-off was further improved by incorporating speech presence information within the vanilla iVector framework. Three different systems were then proposed: (1) the segment-level weighted score fusion system, an ensemble system which incorporated speech presence information at utterance level to weight the classification scores of the baseline and the NF-based system; (2) the frame-level modified Baum-Welch statistics system, where the speech presence information was included at frame-level; and (3) an ensemble system consisting of the late fusion of a modified Baum-Welch statistics system with a complementarily weighted, modified NF-based system. In these systems, the speech presence possibilities were estimated by GMM-based soft voice activity detection (VAD), which can be holistically incorporated into the iVector framework at either the segment level or the frame level. Experiments showed that, for the iVector framework, sufficient background signal information can be extracted during speech pauses to enable a good scene classification. In addition, the NF-based features yield ASC-relevant information even in the speech-dominant frames. By integrating this information, the iVector framework performance can be further improved. The experiments with the iVector framework were conducted on the DCASE2016 dataset, which contains smaller and less complex data (i.e., fewer scenes, devices, cities, ...). However, when it comes to the DCASE2021 dataset, the performance of the iVector framework was less promising, as the baseline accuracy in the absence of foreground speech was only 47.2%, which is only marginally better than the official baseline (45.6%). This demonstrates that the iVector framework cannot generalize well to larger and more complex datasets, thus it is necessary to introduce deep learning into our ASC systems.

However, the comparative analysis in our previous work can still provide extra insights into recent popular ASC systems based on deep learning. In this paper, we

investigate the methods to improve the foreground speech robustness of the deep learning systems with inspiration from the iVector framework.

We refer to [15,16] as complementary reading for this article for a detailed explanation of the iVector framework and noise floor estimate. The chief contributions of these works are summarized in Table 1.

**Table 1.** The chief contributions of our research in previous papers and this paper.

| Paper | System | Contribution |
| --- | --- | --- |
| [15] | iVector | Noise-floor based features; Multi-condition training. |
| [16] | iVector | SoftVAD; Modified Baum-Welch statistics; Score-fusion. |
| This paper | ResNet | Noise-floor based features; CRUSE; Meta-learner. |

## 3. DNN-Based ASC

In this section, we present the ResNet baseline for ASC and multi-condition training. Also, Mel-FilterBank Energies (MFBE) is introduced as the input feature of the baseline.

### 3.1. Model Architecture

In previous work, we proposed [22] a residual network (ResNet) architecture. The architecture of the system is shown in Figures 1 and 2. Except for the input layer, all the pre-pooling layers are Double Conv Blocks which incorporate two convolutional layers, each with a kernel size of $3 \times 3$. The Double Conv Block is detailed in Figure 2. Squeeze-and-Excitation blocks are integrated to leverage global information as well in the frame-level layers. The inclusion of a pooling layer enables the model to handle audio input of varying lengths by computing the temporal mean and standard deviation of input features, creating a fixed-length representation. A final linear layer then maps this fixed-length representation to 10 outputs, representing 10 acoustic scene classes. The activations of these output nodes undergo a softmax layer to calculate the probabilities of different acoustic scenes for each input recording. To implement the softmax layer, we adopt the test configuration of the DCASE2021 development set to train a multinomial logistic regression classifier [24,25] based on the output scores. Note that the ResNet system [22] previously submitted to the DCASE 2021 challenge is either quantized or pruned to meet low-complexity requirements. The model size (non-zero parameters) is 114.0 kibibyte. We consider different model sizes: both small-footprint and large-footprint models by changing the channel sizes (*C1* and *C2* in Figure 1) in the model.

### 3.2. Multi-Condition Training

To improve the robustness of ASC system, multi-condition training [15,18] is performed. This is done consistently for the ResNet baseline and all proposed systems in this paper. Since the ResNet baseline is constructed with the setup of DCASE 2021, the training and test datasets exclusively consist of recordings of the acoustic scene, with no interruption of foreground speech. Multi-condition training with different levels of foreground speech can reduce the mismatch between training and test conditions and improves the robustness of the classification system. In the proposed multi-condition training, in addition to the usage of the original DCASE background noise, a supplementary dataset is created by mixing clean speech with background noise with SBRs randomly chosen from: $[-5, 0, 5, 10, 15, 20]$ dB. This allows the model to learn from different combinations of foreground speech and background noise.

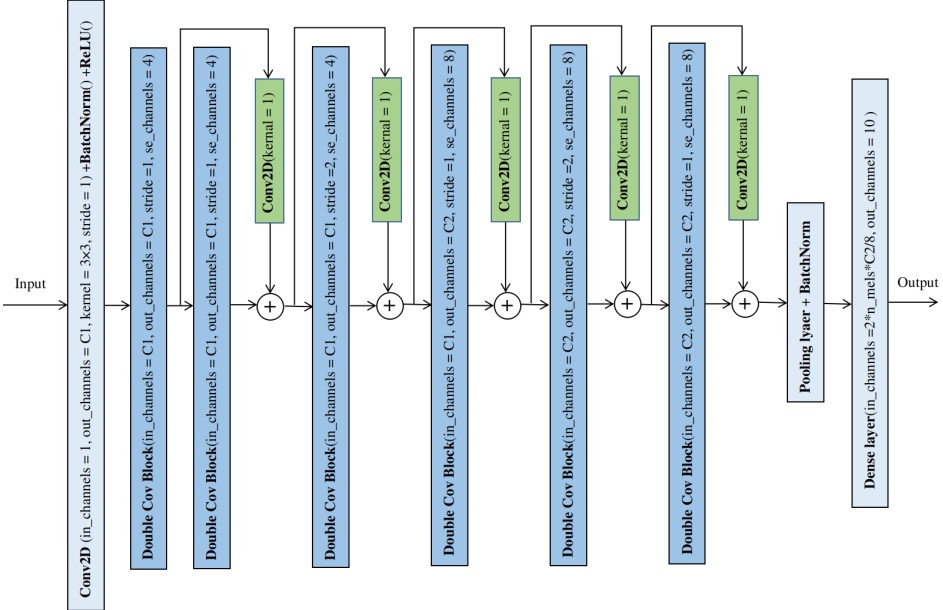

**Figure 1.** ResNet Architecture: C1 = 20 and C2 = 40 in small-footprint model; C1 = 80 and C2 = 160 in large-footprint model.

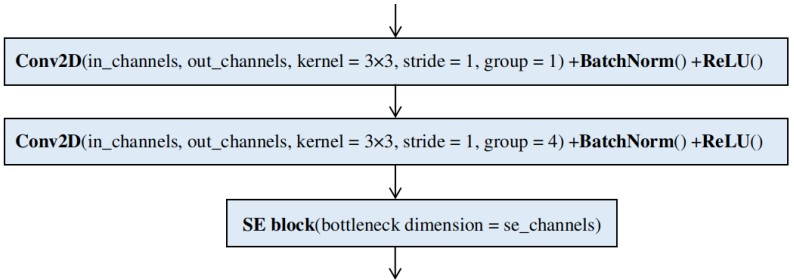

**Figure 2.** Double Conv Block.

### 3.3. Input Features

Before extracting the input features, another data augmentation method, random crop, is applied to address the overfitting issue. The random crop takes a certain fixed time length temporal crop starting on random moments from each training segment. The input feature is then extracted from the temporal crops. In the baseline, the log amplitudes of MFBE of short, overlapping signal segments are adopted. The sampling rate of the input signal is 16 kHz. The time-domain data is first transformed into the Fourier domain after applying a Hann window and computing a 1024-point discrete Fourier transform (DFT). The window size is set to 512 samples and the frame shift is 160 samples. The positive half of the DFT magnitude spectrogram is then mapped by a 128-point Mel-filter bank to the Mel spectrum. Mean normalisation is applied to the log MFBE features of each crop, across time, before they are fed to the neural network.

## 4. Proposed Methods

### 4.1. Noise-Floor-Based Feature for ASC

The input features for the baseline approach are the MBFE features, directly computed on the input signal. Based on this feature, we propose to introduce noise floor estimate as an extra feature into the ASC system to extract information *only* relevant to the background noise. For noise floor, we first calculate an estimate of the power spectral density of the background or ambient signal. It may be obtained by any of the well-known statistical methods (e.g., minimum statistics (MS) methods [19], Minimum Mean-Square Error (MMSE) [21]). Essentially, this consists of recursively smoothing an estimate of the signal periodogram, thereby removing the short variations attributable to speech and preserving

the long-term characteristics of the background. In this work, we adopt Speech Presence Probability Minimum Mean-Square Error (SPP-MMSE) [21] as our noise floor estimation method. Finally, the acoustic features are derived in the same manner as MFBE from the noise floor estimate. We term these the noise-floor-based MFBE features (NF-MFBE).

With this feature, we expect that the background noise information is well preserved while the speech information irrelevant to the acoustic scene is reduced, which should then lead to better classification performance when speech is present. But also as observed in the noise floor application in the iVector framework [15], it unavoidably removes some background noise information and brings in distortion when no speech is present or at very low SBRs. It requires further effort to balance the trade-off so that a consistent improvement is obtained across different SBRs.

### 4.2. Incorporating Speech Presence Information

Alternatively, we consider incorporating speech presence information within the ResNet baseline to extract ASC relevant information only during speech absence. To estimate the speech presence probability, the speech mask from an SE system, CRUSE [23], is provided. The network has a symmetric encoder-decoder structure with added skip connections. The encoder composes of four 2D-convolutional layers for feature extraction, and a grouped-GRU at the bottleneck to efficiently capture the temporal information. The parameters of the network are summarized in Table 2. It is trained on 140 h of the noisy mixture from the DNS challenge 2021 wideband dataset [26] with the signal-approximation loss proposed in [23]. The network takes the noisy amplitude spectrum as input, and estimates a real-valued Time-Frequency (TF) mask to denoise the input signal.

**Table 2.** Parameters of the CRUSE net. Since encoder and decoder have symmetric structures, only the encoder structure is enumerated.

| Encoder Parameters | |
| --- | --- |
| Number of channels | 16, 32, 64, 128 |
| Kernel size (Time, Frequency) | (2, 3) for all layers |
| Stride (Time, Frequency) | (1, 2) for all layers |
| Default activation functions | Leaky ReLU, slope $= 0.03$ |
| Final layer activation functions | sigmoid |
| Skip connection | $1 \times 1$ convolution |
| Number of training epochs | 40 |

A speech mask is utilized as one of the input features which provides speech presence information to the ResNet system. By emphasising the background-dominant frames and reducing the speech-dominant frames, we expect that the ASC-relevant information can be further extracted. The advantage of providing the speech mask as an extra feature is that all information in the recording is still available to the network, without distortion from speech suppression. Thus, we use the speech mask as an extra input feature of the ResNet baseline to let the system learn the best feature extraction in an end-to-end manner from different combinations of background and speech.

### 4.3. Ensemble Methods

The methods in previous sections provide different ASC systems extended from the same ResNet baseline by augmenting the input features. As is evident, the three input features including MFBE, NF-MFBE, and speech mask can be combined in different ways. So, an ablation study is required to determine the most efficient combination. For example, to analyse the influence of NF-based features, three systems can be evaluated: $ResNet_F$ system which uses only the MFBE, $ResNet_{NF}$ system which uses only NF-MFBE, and $ResNet_{F+NF}$ system which uses both MFBE and NF-MFBE. Similarly for evaluating the

benefit of speech presence information, $ResNet_{F+M}$ system which uses both MFBE and speech masks can be benchmarked against $ResNet_F$. Also, with the $ResNet_{F+NF+S}$ system, we expect that the model makes optimal use of all the information available by inherently incorporating the speech presence probability to derive the necessary background signal characteristics from the different features. Each of the combinations is evaluated with small-footprint and large-footprint models.

Ensemble methods [27,28] are often used to reduce classification error by running different classification models in parallel, each of which uses different parameters or different training data. The results of all the models are then combined into a global decision. As we observe, different features yield better discrimination at different SBR ranges. Ensemble methods can combine the test scores of the aforementioned models in a weighted manner to yield an improved final classification result. Therefore, we train a meta-learner consisting of one linear layer followed by a softmax layer, to combine the outputs of the different ResNet configurations. The meta-learner is trained with cross-entropy as loss the validation dataset. Thereby, the output global scores are the weighted combinations of the scores from each model.

For performance evaluation, the meta-learner is also compared with average voting, a simple and effective method that combines the classification results with the averaging weights.

## 5. Results and Discussion

### 5.1. Dataset

The dataset for the different acoustic scenes is taken from TAU Urban Acoustic Scenes 2020 Mobile development dataset [7], which is also used in the DCASE challenge. The 10 acoustic scenes included in the dataset are: airport, shopping_mall, metro_station, street_pedestrian, public_square, street_traffic, tram, bus, metro, and park. The dataset contains data from 10 European cities with a total duration of 64 h. 9 recording devices used in this dataset include 3 real devices and 6 simulated devices. 46 h of data are recorded with real devices, and smaller amounts are from simulated devices where the simulation is done using recorded audio and impulse responses of real devices with additional dynamic range compression. A cross-validation setup is provided for the balanced split in terms of scene classes and devices. The foreground speech dataset is randomly selected from LibriSpeech dataset [29]. To generate scenarios with foreground speech, a mixing dataset is created by adding different clean speech on top of background signal segments with SBRs randomly chosen from $-5$ dB, 0 dB, 5 dB, 10 dB, 15 dB, and 20 dB. The background ASC dataset and mixing dataset are concatenated to form the training dataset and are commonly used to train all proposed ResNet systems. For evaluation, the system performance is evaluated on the testing dataset with and without foreground speech. We make sure there is no overlap between the speaker set used for mixing the training data and the speaker set used on the evaluation data.

To measure the reliability of classification accuracy, different cross-validation setups are used and the accuracy is presented in terms of 90% confidence intervals (CI).

### 5.2. Training Setup

We apply random crop as data augmentation, which randomly takes a 3 s temporal crop from each training segment. For NF-based features, to make sure the noise floor estimate converges, we take a 5 s crop and discard the first 2 s for fair comparisons. During training, we optimize the cross-entropy loss using the Adam optimiser. We implement a Cyclical Learning Rate (CLR) schedule and conduct three complete cycles employing the triangular2 policy [30]. The maximum and minimum learning rates are set at $10^{-3}$ and $10^{-6}$, respectively. The maximum learning rate undergoes a decay of a factor of two after each full cycle, which consists of 40 epochs. Additionally, a weight decay value of $10^{-4}$ is imposed. For the training of the meta-learner, we utilize the cross-entropy loss along

with the Adam optimiser, applying a learning rate of $10^{-3}$. For training the SE system, the parameters are summarized in Table 2.

### 5.3. Performance Evaluation

Five systems are trained based on these settings for each complexity:

1.  $ResNet_F$: the ResNet baseline system;
2.  $ResNet_{NF}$: the ResNet system with NF-MFBE;
3.  $ResNet_{F+NF}$: the ResNet system with both MFBE and NF-MFBE;
4.  $ResNet_{F+M}$: the ResNet system with both MFBE and speech mask;
5.  $ResNet_{F+NF+M}$: the ResNet system with MFBE, NF-MFBE, and speech mask.

Based on different combinations of these systems, the performance of average voting and meta-learner is evaluated. The classification accuracies of the mentioned systems are presented in Table 3 with the small-footprint model and Table 4 with the large-footprint model. Each column represents a test condition with a fixed SBR of the testing dataset ('*Clean*' represents the testing condition where no foreground speech is presented).

**Table 3.** Classification accuracy (in %) of different systems, along with the 90% confidence interval. These are results obtained for the small footprint model across the different tested SBRs.

| Accuracy (%) | Clean | −5 dB | 0 dB | 5 dB | 10 dB | 15 dB | 20 dB |
|---|---|---|---|---|---|---|---|
| $ResNet_F$ | 66.66 ± 0.74 | 62.35 ± 0.40 | 61.64 ± 0.58 | 57.86 ± 0.44 | 54.24 ± 0.93 | 49.71 ± 0.67 | 46.39 ± 0.69 |
| $ResNet_{NF}$ | 63.40 ± 0.64 | 57.69 ± 0.71 | 56.97 ± 0.71 | 55.91 ± 0.95 | 52.35 ± 1.10 | 47.97 ± 0.83 | 41.50 ± 0.87 |
| $ResNet_{F+NF}$ | 65.34 ± 1.14 | 61.57 ± 1.08 | 60.61 ± 0.72 | 59.11 ± 0.85 | 55.62 ± 1.49 | 52.78 ± 0.78 | 47.92 ± 1.15 |
| NF_vote_1 | 66.93 ± 0.35 | 62.80 ± 0.20 | 62.90 ± 0.56 | 59.92 ± 0.72 | 55.87 ± 0.35 | 51.39 ± 0.58 | 47.10 ± 0.58 |
| NF_vote_2 | 68.17 ± 0.75 | 64.53 ± 0.67 | 63.19 ± 0.66 | 60.25 ± 0.53 | 56.70 ± 0.50 | 53.18 ± 0.48 | 48.71 ± 0.36 |
| $ResNet_{F+M}$ | 63.07 ± 0.32 | 60.15 ± 0.96 | 60.04 ± 1.21 | 59.01 ± 0.92 | 56.56 ± 0.58 | 54.56 ± 0.96 | 52.18 ± 1.10 |
| M_vote | 68.66 ± 0.41 | 66.02 ± 0.46 | 64.35 ± 0.49 | 61.65 ± 0.55 | 58.90 ± 0.70 | 56.45 ± 0.77 | 54.19 ± 1.77 |
| $ResNet_{F+NF+M}$ | 55.06 ± 1.46 | 56.36 ± 1.52 | 56.09 ± 1.59 | 56.52 ± 1.71 | 55.24 ± 1.66 | 53.21 ± 1.54 | 49.55 ± 1.64 |
| M_NF_vote | 70.48 ± 0.43 | 66.69 ± 1.03 | 65.76 ± 1.58 | 63.29 ± 1.37 | 59.90 ± 1.07 | 56.68 ± 1.00 | 53.80 ± 0.71 |
| Meta_learner | **70.94** ± 0.58 | **67.37** ± 0.92 | **66.29** ± 1.44 | **64.25** ± 1.11 | **61.43** ± 0.95 | **58.60** ± 0.89 | **55.50** ± 1.19 |

**Table 4.** Classification accuracy (in %) of different systems, along with the 90% confidence interval. These are results obtained for the large footprint model across the different tested SBRs.

| Accuracy (%) | Clean | −5 dB | 0 dB | 5 dB | 10 dB | 15 dB | 20 dB |
|---|---|---|---|---|---|---|---|
| $ResNet_F$ | 71.70 ± 0.53 | 66.99 ± 1.10 | 66.31 ± 1.12 | 64.79 ± 0.98 | 62.14 ± 0.87 | 58.50 ± 1.30 | 53.14 ± 1.54 |
| $ResNet_{NF}$ | 67.81 ± 0.57 | 61.45 ± 1.34 | 61.33 ± 1.88 | 60.34 ± 1.34 | 57.25 ± 1.67 | 53.38 ± 1.49 | 48.73 ± 1.97 |
| $ResNet_{F+NF}$ | 67.23 ± 1.34 | 61.33 ± 1.18 | 61.67 ± 0.91 | 60.85 ± 1.34 | 56.87 ± 1.48 | 53.47 ± 1.54 | 49.54 ± 1.48 |
| NF_vote_1 | 72.72 ± 0.31 | 67.32 ± 0.68 | 66.59 ± 1.54 | 65.28 ± 1.10 | 62.66 ± 1.22 | 59.75 ± 1.33 | 54.71 ± 1.25 |
| NF_vote_2 | 73.16 ± 0.07 | 67.52 ± 1.07 | 67.75 ± 1.33 | 65.61 ± 0.91 | 62.68 ± 1.06 | 59.06 ± 1.08 | 55.18 ± 0.90 |
| $ResNet_{F+M}$ | 70.82 ± 0.79 | 66.69 ± 0.75 | 66.66 ± 0.48 | 65.51 ± 0.84 | 64.41 ± 1.46 | 62.56 ± 1.72 | 59.26 ± 1.80 |
| M_vote | **75.33** ± 0.61 | 70.96 ± 0.87 | 70.47 ± 1.01 | 68.80 ± 0.99 | 67.15 ± 0.66 | 64.45 ± 0.50 | 60.89 ± 0.88 |
| $ResNet_{F+NF+M}$ | 64.90 ± 1.09 | 62.63 ± 1.42 | 62.90 ± 1.96 | 62.19 ± 1.90 | 60.52 ± 1.74 | 59.56 ± 1.36 | 56.61 ± 1.05 |
| M_NF_vote | 74.73 ± 0.51 | 70.00 ± 0.91 | 69.99 ± 1.06 | 68.90 ± 1.18 | 67.57 ± 0.77 | 64.38 ± 0.72 | 60.36 ± 1.09 |
| Meta_learner | 74.91 ± 0.66 | **71.43** ± 0.82 | **70.65** ± 0.86 | **69.64** ± 1.07 | **67.65** ± 0.99 | **65.81** ± 0.61 | **61.94** ± 1.17 |

#### 5.3.1. Performance of Small Footprint Models

To enable a fair comparison, we start from a competitive ResNet baseline. Note that the iVector framework in previous work *only* obtains an accuracy of 47.2% in clean conditions with the DCASE 2021 dataset, thus is not suitable for further investigation. The ResNet model we proposed in [22] achieved an accuracy of 68.8% in DCASE 2021 Task 1. Here, models without quantisation or other constraints on memory are considered, to decouple these limiting effects when benchmarking the benefits of the proposed modifications. In the first row of Table 3, this ResNet model (C1 = 20, C2 = 40) is considered as the baseline. With

the MCT dataset, $ResNet_F$ system achieves reasonable accuracy in clean conditions, while the performance deteriorates as SBRs increase. For high SBR (20 dB), there is a significant degradation of approximately 20%. Similar to our observation in the iVector framework, the ResNet model is also not robust against the presence of foreground speech.

Next, $ResNet_{NF}$ that uses NF-MFBE as the input feature is tested. The rest of the system is kept the same as $ResNet_F$. For all tested SBRs, simply replacing MFBE with NF-MFBE as an input feature does not bring any benefits, and, contrary to what we observed in the case of the iVector system, degrades the classification performance. Since the noise floor is obtained by recursively smoothing the signal periodogram, the background signal features can be distorted. Because the estimate is an extremely smoothed signal spectrum with less spectro-temporal variation, this either leads to overfitting or loss of discriminative information for the classification.Next, we consider different combinations of the NF-MFBE and MFBE. In $ResNet_{F+NF}$ (the third row of Table 3), the NF-MFBE and MFBE are both used as the input features. This yields marginal improvements at high SBRs and shows slight degradation in low SBR conditions (below 0 dB). Further, two voting systems are also presented in $NF\_vote\_1$ and $NF\_vote\_2$: $NF\_vote\_1$ is the average voting of $ResNet_F$ and $ResNet_{NF}$, and $NF\_vote\_2$ is the average voting of $ResNet_F$ and $ResNet_{F+NF}$. We expect a balanced performance with voting systems in different scenes. In both of the systems, the improvement is consistent in all SBRs, and $NF\_vote\_2$ achieves the best performance. The results demonstrate that the NF-based feature does contain information relevant to ASC and integrating this information can further improve the ASC system performance in the small footprint model.

Another method we consider is to incorporate the speech presence information in order to treat speech-dominant and noise-dominant frames with different importance, giving more focus to background-dominant frames and less focus to speech-dominant frames. In $ResNet_{F+M}$, the speech presence probability obtained from CRUSE is utilized as an extra input feature of the model. The accuracy of $ResNet_{F+M}$ in high SBRs (from 5 dB) is higher than the $ResNet_F$ baseline, which means that sufficient ASC information can already be extracted from the background-dominant frames in each segment. However, in clean or low SBR conditions, the performance of $ResNet_{F+M}$ is worse. This could be because the speech presence probability introduces more distortion in low SBR conditions, due to inaccurate estimates, thereby losing ASC-relevant information. Due to the different performances of $ResNet_F$ and $ResNet_{F+M}$, $M\_vote$ that averages scores of $ResNet_F$ and $ResNet_{F+M}$ is introduced. This system shows competitive performance in both low and high SBRs.

Based on the results of the NF-based features and speech mask, $ResNet_{F+NF+M}$ could be a better solution, which is expected to weight different kinds of features adaptively in different frames. Specifically, for background-dominant frames, NF-MFBE features can be adopted and MFBE can be adopted in speech-dominant frames. Surprisingly, however, the results indicate that we do not obtain better performance by incorporating more features as network input. Especially in low SBRs, the accuracy is even worse than both $ResNet_{F+NF}$ and $ResNet_{F+M}$. A possible explanation is that both NF-MFBE features and speech mask lose/distort information in low SBR conditions and then add up to further degradation. To take full advantage of the trained networks, we propose two alternatives to tackle this problem: $M\_NF\_vote$ and $Meta\_learner$. The first alternative $M\_NF\_vote$ is the average voting by $ResNet_F$, $ResNet_{F+M}$ and $ResNet_{F+NF}$. It shows competitive improvement in all SBRs. The second alternative, the meta-learner, uses a fully connected layer for the combination of three models shows the largest improvement. The comparison of the $ResNet_F$ baseline and $Meta\_learner$ is presented in Figure 3. At 20 dB SBR, an absolute improvement of 9.11% is observed.

For the small-footprint model, the following key points can be concluded:

* NF-based features bring benefits to certain acoustic scenes, thus can be utilized as an extra input feature for ASC;

* Incorporating speech masks improves ASC accuracy in high SBRs, while losing information in low SBRs;
* Ensemble methods in general help ASC and the best performance can be obtained by *Meta_learner* that combines models with three different input features.

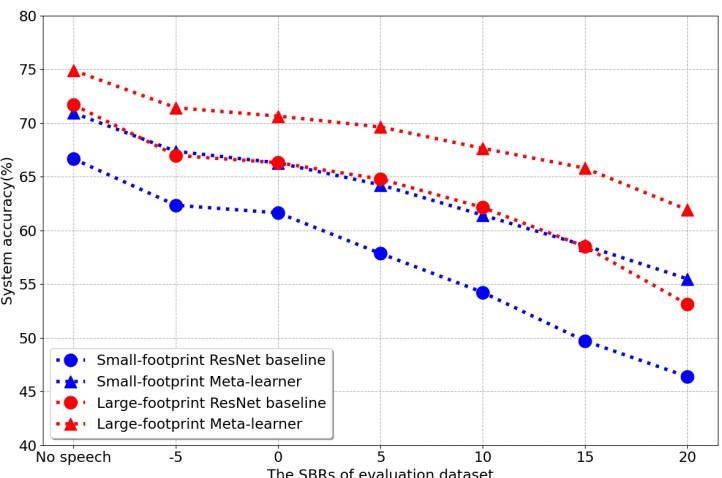

**Figure 3.** The performance of both small- and large-footprint ResNet baseline and the proposed best systems.

### 5.3.2. Performance of Large-Footprint Model

Table 4 presents performances of the extensions based on the large-footprint model. Comparing the $ResNet_F$ baseline with the small- and large-footprint, we found that the large model extracts more ASC information and improves the accuracy of all tested conditions. This conclusion also holds for other extensions. In other words, increasing model size is a consistent way of improving classification accuracy. When extending the $ResNet_F$ baseline with NF-MFBE features, it is surprising that the small- and large-footprint model behave differently. In the large model, though $ResNet_{NF+F}$ still outperforms the corresponding system in the small model, it shows poorer performance than the large $ResNet_F$ baseline. We do not find evidence indicating that NF-based features can be beneficial to ASC in the large-footprint model. The reason might be that sufficient ASC information can already be extracted from MFBE features with a larger model, and no extra benefits could be brought by noise floor estimate in this condition. Model voting with NF-based features could still make slight improvements in $NF\_vote\_1$ and $NF\_vote\_2$, but it could also be attributed to the increase of system complexity.

By incorporating speech presence information from cruse-net in the large-footprint model, a significant improvement is obtained by $ResNet_{F+M}$ in almost all conditions, except a very slight 'decrease' in the clean condition. $M\_vote$ further improves the system performance. For the combination of three features in the large model, $ResNet_{F+NF+M}$ tends to bring larger distortion than $ResNet_{F+M}$ in all SBRs. $M\_NF\_vote$ could still improve the system performance compared with the $ResNet_F$ baseline. It is interesting to note that $M\_vote$ and $M\_NF\_vote$ show almost identical performance, indicating again that no additional information may be attributed to the NF-based features. To conclude, both $ResNet_{F+NF+M}$ and $M\_NF\_vote$ show that noise floor estimate does not bring further improvement to the large-footprint model. The best performance is achieved by the *Meta_learner* for the large model. The absolute improvement is 8.8% in 20 dB conditions. For the large-footprint model, the following key points can be concluded:

* Statistical noise floor estimate does not convincingly help ASC, presumably because the model is big enough that relevant information can be faithfully extracted from the feature of mixed signal;
* For the large model, incorporating DL-based speech mask estimates consistently improves the classification accuracy;

∗   Ensemble methods could further bring benefits to the large-footprint model.

### 5.3.3. Complexity Considerations

In the previous session, the comparisons were made based on the small- and large-footprint model. However, ensemble methods that combine different models increase the system complexity. It is important for a fair comparison to be made with the knowledge of system complexities. In Table 5, the system complexity of all the extensions of the small-footprint model and the large-footprint model is displayed. When considering ensemble methods, *NF_vote*_1, *NF_vote*_2, and *M_vote* double the model size while *M_NF_vote* and *Meta_learner* triple the model size. Here, we do not consider the complexity of cruse-net since it can also be replaced by other speech enhancement systems.

**Table 5.** System complexities in terms of non-zero parameters in kilobyte (KB) and Multiply-Accumulate Operations (MACs).

| Non-Zero Parameters/MACs | Small-Footprint Systems | Large-Footprint Systems |
|:---:|:---:|:---:|
| $ResNet_F$ | 336 KB/1.58 G | 4450 KB/24.72 G |
| $ResNet_{NF}$ | 336 KB/1.58 G | 4450 KB/24.72 G |
| $ResNet_{F+NF}$ | 336 KB/1.58 G | 4450 KB/24.72 G |
| NF_vote_1 | 672 KB /3.16 G | 8900 KB/49.44 G |
| NF_vote_2 | 672 KB /3.16 G | 8900 KB /49.44 G |
| $ResNet_{F+M}$ | 336 KB/1.58 G | 4450 KB/24.72 G |
| M_vote | 672 KB/3.16 G | 8900 KB/49.44 G |
| $ResNet_{F+NF+M}$ | 336 KB/1.58 G | 4450 KB/24.72 G |
| M_NF_vote | 1008 KB/4.74 G | 13350 KB/74.16 G |
| Meta_learner | 1008 KB/4.74 G | 13350 KB/74.16 G |

In Figure 3, the performances of both small- and large-footprint $ResNet_F$ baseline and the corresponding best systems are plotted. It can be observed that the proposed methods improve the foreground speech robustness for both the small- and large-footprint models. It should be noted that the small-footprint *Meta_learner* shows similar performance to the large-footprint $ResNet_F$ baseline, which indicates that though relatively compact, the proposed *Meta_learner* can efficiently fuse different models.

### 6. Conclusions

To improve the foreground speech robustness of DL-based ASC systems, we proposed and evaluated several methods building on insights from previous work on a stochastic classification model for ASC. We systematically investigated the benefits of incorporating noise floor and speech presence probability estimates within a contemporary ResNet-based DNN-framework. Firstly, the experimental study showed that NF-MFBE improves system performance at high SBRs without increasing the complexity of the small-footprint model. In high-complexity architectures, when ASC-relevant information could already be sufficiently extracted from the original MFBE feature, the NF-based feature is then redundant, even degrading the performance. In contrast, by incorporating the speech presence probability, the system performance is significantly improved for both small- and large-footprint models. Owing to a better estimation of speech presence probability at the time-frequency scale, the ASC system can pay more attention to the background-dominant frames to produce more reliable classification results. Next, Ensemble methods bring extra benefit by combining the scores of different systems into one consistent decision, but this improvement in performance comes at the cost of increasing the system complexity. Aiming at reducing the effect of foreground speech, at the SBR of 20 dB, we achieve an absolute improvement in classification accuracy of 9.1% for the small-footprint model and 8.8% for large-footprint model. This investigation can provide a reference for choices in real applications, where the balance between system complexity and performance must be considered.

**Author Contributions:** Conceptualization, N.M.; Data curation, S.S.; Investigation, S.S. and Y.S.; Methodology, S.S. and Y.S.; Software, S.S. and Y.S.; Supervision, N.M.; Validation, S.S.; Writing—original draft, S.S.; Writing—review & editing, Y.S. and N.M. All authors have read and agreed to the published version of the manuscript.

**Funding:** This research received no external funding.

**Institutional Review Board Statement:** Not applicable.

**Informed Consent Statement:** Not applicable.

**Data Availability Statement:** Data is contained within the article.

**Conflicts of Interest:** The authors declare no conflict of interest.

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
