# Peer review of "Robust Detection of Background Acoustic Scene in the Presence of Foreground Speech"

_applsci, doi:10.3390/app14020609_

Round 1
Reviewer 1 Report
Comments and Suggestions for Authors
The authors have extended their previous work to detect background acoustic science when speech is presence. The paper is well written and clearly explain their new contributions.
One criticism is that the mere is no description of detected acoustic scenes other than the name of the data base. The paper will get increased readership if what kind of backgrounds are in these scenes are explained. Currently, the evaluation and discussion is too mechanical. For the final version, please include a paragraph on this for the benefit of wide research community.
Author Response
Thank you very much for taking the time to review this manuscript. Please find the corresponding corrections highlighted in the re-submitted files. The classes of acoustic scene are added to the paper.
Reviewer 2 Report
Comments and Suggestions for Authors
This manuscript aims to improve the foreground speech robustness through extending the different strategies to the deep-learning (DL) -based Acoustic Scene Classification (ASC) systems, and the optimal results are validated by the experimental study. Some corrections are suggested to make manuscript more reasonable and easier to understand.
1. In the abstract, it will be better to add some quantitative data instead of qualitative discussions, which can make readers more directly understand the main achievements of this research. The authors can select the most important results from the conclusion section to enrich the abstract.
2. This manuscript refer to the references [15] and [16] as the complementary read to this article for a detailed explanation of the iVector framework and noise floor estimate. It would be better to make a table to compare the utilized methods and results in the 2 references and the major improvement in this article, which is more perspicuous.
3. It could be judged from the Tables 2 and 3 that the improvements of classification accuracies for the various testing conditions are different, please explain the possible reason for this phenomenon.
4. In the section 5.3.3, is there any other index to shown the system complexities except the non-zero parameters in kilobytes? It would be better to exhibit the novelty of the proposed algorithm in this manuscript from more perspectives.
5. In the conclusion section, it would be better to conclude the major achievements in this manuscript into several points and list them one by one, which can make the readers understand easily. Meanwhile, it would be better to show some quantitative results in the conclusion, which is more convincing.
Author Response
Thank you very much for taking the time to review this manuscript. Please find the detailed responses below and the corresponding corrections highlighted in the re-submitted files.
- Quantitative results are added to both the abstract and conclusion.
- The table is added in the corresponding section.
- In section 5.3, we mainly explained the reason for achieving different performance in different scenarios. Within the same system, in different SBR conditions, indeed the system performs differently. This is because, in general, the stronger the foreground speech is, the more ASC-relevant information is lost. However, this is not linear. Therefore, the improvements in classification accuracies are different.
- Another way of showing system complexity, the MACs (multiply and accumulate operations), is added to the table.
- The conclusion is reorganised and quantitative data are added as suggested.
Reviewer 3 Report
Comments and Suggestions for Authors
Based on the vector framework, the authors propose different strategies to improve the classification accuracy when foreground speech is present, and extend these methods to deep learning (DL) -based ASC systems to improve the robustness of foreground speech. The X experimental study systematically validated the contribution of each proposed modification, and the results showed that the classification accuracy of all tested SBRS was improved using the proposed input features and meta-learners. In addition; Under the condition of low SNR; The practicability of this method needs to be supplemented
Author Response
Thank you very much for taking the time to review this manuscript. Please find the corresponding corrections highlighted in the re-submitted files.
Since the main aim of the research is to improve the ASC robustness in the presence of foreground speech, we mainly consider the relatively high SBR conditions – which are the most challenging conditions for scene classification. For low SBR conditions, the situation is less critical, since the background signal is more dominant – allowing for easier scene classification. Even in these cases, however, the proposed system still shows a slight advantage. Thus, it is a practical system for both low and high SBRs.
Reviewer 4 Report
Comments and Suggestions for Authors
line 43 Gaussian mixture models (GMMs) [2], hidden Markov models (HMMs) [3], and iVector [4,5].
Please explain briefly the characteristics of each model here.
Author Response
Thank you very much for taking the time to review this manuscript. Please find the corresponding corrections highlighted in the re-submitted files.
The characteristics of each model are added in the corresponding section.